# From Organelle Morphology to Whole-Plant Phenotyping: A Phenotypic Detection Method Based on Deep Learning

**DOI:** 10.3390/plants13091177

**Published:** 2024-04-23

**Authors:** Hang Liu, Hongfei Zhu, Fei Liu, Limiao Deng, Guangxia Wu, Zhongzhi Han, Longgang Zhao

**Affiliations:** 1College of Grassland Science, Qingdao Agricultural University, Qingdao 266109, China; lhfiread@163.com; 2College of Computer Science and Technology, Tiangong University, Tianjin 300387, China; zhuhongfei98@163.com; 3College of Science and Information, Qingdao Agricultural University, Qingdao 266109, China; feiliu5669@sina.com (F.L.); denglm68@163.com (L.D.); 4College of Agronomy, Qingdao Agricultural University, Qingdao 266109, China; wuguangxia2020@163.com

**Keywords:** organelle, plant phenotypes, deep learning, *Arabidopsis thaliana*, breeding

## Abstract

The analysis of plant phenotype parameters is closely related to breeding, so plant phenotype research has strong practical significance. This paper used deep learning to classify *Arabidopsis thaliana* from the macro (plant) to the micro level (organelle). First, the multi-output model identifies Arabidopsis accession lines and regression to predict Arabidopsis’s 22-day growth status. The experimental results showed that the model had excellent performance in identifying Arabidopsis lines, and the model’s classification accuracy was 99.92%. The model also had good performance in predicting plant growth status, and the regression prediction of the model root mean square error (RMSE) was 1.536. Next, a new dataset was obtained by increasing the time interval of Arabidopsis images, and the model’s performance was verified at different time intervals. Finally, the model was applied to classify Arabidopsis organelles to verify the model’s generalizability. Research suggested that deep learning will broaden plant phenotype detection methods. Furthermore, this method will facilitate the design and development of a high-throughput information collection platform for plant phenotypes.

## 1. Introduction

Plants are an essential material resource for human survival [1]. The plant phenotype refers to the observable structural, functional, and morphological characteristics, and their changing laws under specific circumstances [2]. Plant phenotyping has solved many related problems in recent years, and image analysis systems have been widely used in phenotyping research [3]. Methods based on image analysis can measure and monitor plant growth and perform morphological analyses. Furthermore, these methods are able to use phenotype extraction to analyze the plant’s different organ images and further analyze the traits of plants to predict yield [4]. Plant phenotype-related research is significant in food security and environmentally sustainable development.

Traditional plant identification and classification methods mostly rely on expert experience to analyze the morphological phenotypes of plants, such as appearance, texture, and color. However, these features are subjective and inaccurate [5]. To improve the production control and intelligent management level of smart agriculture, more powerful phenotypic data analysis methods are needed [6]. Compared with manual detection, machine vision technology effectively allows the non-destructive and high-throughput acquisition of crop phenotypes [7]. The high-throughput phenotype analysis platform based on digital image technology is becoming more and more perfect [8]. Machine vision technology was used in agroforestry production practice aspects, especially plant phenotype research.

Image processing was widely utilized in plant phenotyping. Gebhardt et al. [9] proposed a grass weed leaf segmentation algorithm, and the residual grass leaves in the binary image were removed using morphological opening. Abbasgholipour et al. [10] developed a genetic algorithm-based sorting system for raisin segmentation. The study aimed to develop a machine vision-based raisin detection technique suitable for various lighting conditions. They used a permission-coded genetic algorithm (GA) to identify regions in the hue–saturation–intensity (HSI) color space (GAHSI) for supervised segmentation of color images. GAHSI provided a reliable method for separating raisin areas. However, the performance of image processing methods was not stable. This instability may be due to the effect of variable conditions on the segmentation results, and the article mentions that the experiments require further studies on various imaging devices, color conversions, GA coding, and operators. Furthermore, there is no significant difference in the GAHSI method compared to the segmentation results based on cluster analysis. This implies that the GAHSI method may not always outperform other segmentation methods, further suggesting that the performance of this image processing method is unstable. This suggests that the current method may not be fully optimized and may not perform well under different lighting conditions or when using different types of raisins.

Machine learning methods have applications in plant phenotyping. Azim et al. [11] proposed an improved K-Nearest Neighbor (KNN) algorithm based on K-means to detect disease spots on rice leaves. This study reported an overall accuracy of over 94% under field conditions. Saleem et al. [12] compared different classifiers’ performance in detecting plant leaf species. The KNN performance was the best, and the recognition accuracy was 97.6%. The machine learning methods mentioned above needed to select and extract image characteristic information, and the operation process was relatively cumbersome.

With the advances of deep learning in feature extraction, many artificial intelligence algorithms have shown high accuracy and robustness in identifying plant-specific morphological structures, providing a robust solution for the intelligent identification of plant phenotypes. Giuffrida et al. [13] proposed a unified and general deep learning architecture for Arabidopsis leaf counting that can count plant leaves from visible, fluorescence, and near-infrared 2D images. Lim et al. [14] analyzed Arabidopsis pollen tetrads using DeepTetrad, a deep learning-based image recognition package. Kolhar et al. [15] used a combination of CNN and ConvLSTM to identify spatially and temporally characterized Arabidopsis lines. CNN–ConvLSTM achieved 97.97% accuracy with very few trainable parameters. Mishra et al. [16] used a combined deep learning and chemometrics approach for processing close-range spectral images to detect drought stress in *Arabidopsis thaliana* to support digital phenotyping. Xu et al. [17] proposed a localization and classification method based on a two-level variable domain fusion network to detect tea sprout detection. The method’s detection accuracy of tea buds reached 95.71%.

Most current deep learning models are single-task networks, which means many models only learn one task at a time [18]. However, more complex situations need to be considered in the application process, so using a single-task network cannot meet the application requirements. Multi-task learning can improve model accuracy and generalization performance, especially when dealing with multiple related tasks [19]. Dobrescu et al. [20] proposed a multi-task deep learning framework for plant phenotyping to infer leaf number, projected leaf area, and genotype classification. Wang et al. [21] proposed a multi-task model for plant diseases to detect plant species and the accuracy of disease detection was 84.71% and 75.06%.

Deep learning has shown great potential in plant phenotyping research. Deep learning techniques are used to contribute to breeding programs by identifying Arabidopsis genotypic lines and predicting plant growth states, which are key factors in plant breeding. Deep learning techniques can facilitate the development of high-throughput information collection platforms for plant phenotyping, thus providing efficient data collection and analysis for plant research. The abovementioned multi-task learning studies mainly focus on multiple classification problems, so they still have limitations in the constructed model. These limitations mainly come from the following aspects: In terms of data requirements, a large number of samples are needed to ensure the generalization performance of the model. In terms of computational resources, a high-performance graphics processing unit (GPU) or tensor processing unit (TPU) is required to ensure the construction and training of large deep learning models. In terms of hyperparameters, such as learning rate, batch size, and number of layers, choosing the appropriate hyperparameter configuration is crucial for the performance of the model. It is also important for the migration learning capability of the model. In multi-task learning, there may be differences among tasks, which can affect the effectiveness of migration learning. The success of transfer learning relies on the correlation among tasks; if the correlation is low, the model may have difficulty in learning useful information from one task and applying it to another.

As a model plant, *Arabidopsis thaliana* is widely used in plant genetics, cell biology, molecular biology, and population chemistry. Arvidsson et al. [22] described an automated growth phenotyping process for *Arabidopsis thaliana* that integrates image analysis and rosette region modeling to precisely quantify genotypic effects. A non-invasive automated phenotyping method for small plants such as *Arabidopsis thaliana* is described, which can significantly improve the efficiency and accuracy of phenotyping. The system can not only rapidly detect significant growth phenotypes, but even identify weaker phenotypes, which has important implications for functional genomics and systems biology research. Dhondt et al. [23] describes a new in vitro imaging system (IGIS) for real-time analysis of Arabidopsis inflorescence growth. Arabidopsis plants are grown in petri dishes placed on rotating plates, and the system tracks the growth of the plants by automatically taking images every hour. Not only can the growth dynamics of *Arabidopsis thaliana* be monitored and analyzed, but the growth phenotypes under different stresses can also be observed in a time-resolved way. Vasseur et al. [24] combines the automated plant image computerization of ImageJ with subsequent statistical modeling in the R language. This method provides plant biologists with an easy tool to measure the growth dynamics and fruit count of hundreds of individuals in simple computational steps. This study demonstrates that the method of estimating fruit number from inflorescence images not only effectively predicts fruit number, but also correlates it with growth variation through image analysis to help understand plant performance and adaptation under different genotypes and environmental conditions.

In this paper, we propose a new multi-task learning method, which can not only complete the classification task but also have the function of regression prediction. In addition, this multi-output model can modify the output of the model according to specific task requirements. As a model plant, *Arabidopsis thaliana* is widely used in plant genetics, cell biology, molecular biology, and population chemistry. In this study, *Arabidopsis thaliana* was systematically classified from the macro to micro level, and the analysis from plant individual to suborganelle was realized. Arabidopsis is an important model plant in basic biology. In biology, the structural hierarchy of living systems is from cells, tissues, organs, systems, individuals, populations and communities, ecosystems, and biospheres. In order to ensure the consistency and rigor of the experiment, the suborganelles of Arabidopsis plants were selected for the transfer learning test in the subsequent experiment. The specific research contents are as follows:

Deep learning was used to classify *Arabidopsis thaliana* from the macro (plant) to micro (organelle) level.A multi-task model was used to study plant phenotypes (*Arabidopsis thaliana*), complete the classification of *Arabidopsis thaliana* genotypes, and predict the growth state of the plant.Arabidopsis organelles (chloroplast, mitochondria and peroxisome) were identified using an improved multitasking model to test the migratory nature of the method.

## 2. Materials and Methods

The workflow of the multi-output network for Arabidopsis image recognition is shown in Figure 1. Firstly, the multi-output model was used to recognize and predict the background of *Arabidopsis thaliana* temporal images. Then, the background of *Arabidopsis thaliana* images was removed by image preprocessing, and the processed images were imported into another multi-output model for training. This experiment aimed to research the effect of image background on *Arabidopsis thaliana* recognition. Then, the time-series images of Arabidopsis were collected at intervals and imported into the model for training. This experiment aimed to research the effect of time intervals on model performance. Finally, the multi-task model was modified to classify the organelle images of *Arabidopsis thaliana*. Classification is the process of accurately identifying Arabidopsis genotypic lines and predicting plant growth states, which involves training the model to analyze and interpret complex data patterns to achieve accurate classification. The results presented in the following sections prove the model’s robustness.

### 2.1. Materials

The Arabidopsis dataset was derived from the Arabidopsis data collected by Namin et al. [25]. The data included four Arabidopsis accessions, accession Sf-2 (Sf-2), accession Cvi (Cvi), Landsberg (Ler-1), and Columbia (Col-0). The number of images in each category is shown in Table 1. Arabidopsis plants were placed in an image acquisition and growth chamber, and a camera was installed above the growth chamber to monitor the plants. The growth status of *Arabidopsis thaliana* was recorded at 12:00 noon every day. *Arabidopsis thaliana* plants were collected for 22 consecutive days in the experiment, so each plant contained time-series image data for 22 days. A preprocessing step was applied to the captured image before the classification task. First, to eliminate the effect of image distortion, we used the color card pasted indoors to correct the image’s color and finally make the captured image similar to the color of real plants. A time-matching method was then used to detect trays and pots in the growth chamber to extract images of each pot to generate an image sequence of each plant’s growth.

To verify the generalizability of the proposed method, we also applied the model to identify Arabidopsis organelles [26]. The Arabidopsis organelle dataset was obtained from Arabidopsis with the col-0 genotype. The dataset contained images of three Arabidopsis organelles, including chloroplast, mitochondria, and peroxisome. Imaging was performed on the cotyledons of 7–10 day-old seedlings using an Olympus Spectral FV1000 laser scanning confocal microscope through a 60× oil objective (N.A. = 1.42). Chloroplasts were imaged using chlorophyll autofluorescence by laser excitation at 488 nm and collecting emission spectra at 665–700 nm. Mitochondrial and peroxisome imaging, on the other hand, utilized the yellow fluorescent protein (YFP) labeling by laser excitation at 515 nm wavelength and emission spectra collected at 530–580 nm wavelength. All images were acquired at 1024 × 1024 pixel resolution, pseudo-stained yellow for analysis, and saved as JPEG files. The detailed information of the image is shown in Table 2. In the study, normal and damaged division of chloroplasts, mitochondria, and peroxisomes of *Arabidopsis thaliana* (Col-0) plants were observed by laser confocal scanning microscopy in order to apply deep learning tools for organelle image analysis. Normal chloroplast images were collected from wild-type plants, whereas images of damaged chloroplasts were obtained from the mutant arc6-5, which affects the accumulation and replication of chloroplasts 6 (ARC6), a key protein in chloroplast division. Normal mitochondrial images were obtained by transferring ScCOX4 with a yellow fluorescent protein (YFP) marker into wild-type plants, whereas damaged mitochondrial images were obtained from the mutant drp3A-2, which affects the dynamin-related protein 3A of mitochondrial and peroxisomal division. Normal peroxisomal images were acquired from wild-type plants transformed with the YFP-PTS1 marker. PTS1-tagged wild-type plants and abnormal peroxisome images were obtained from plants overexpressing YFP-PEX11d.

### 2.2. CNN

The essence of the convolutional neural network is a multi-level perceptron, which directly maps nonlinear inputs to outputs [27]. Its main features are weight sharing and local linking, which can greatly reduce the number of weights and the complexity of model acquisition [28]. The convolutional neural network can directly extract the original image data as input, avoiding the complex data image extraction pretreatment in the early stage [29]. Convolutional neural networks are usually composed of an input layer, a full connection layer, a pooling layer, a convolutional layer, and an activation layer. The hierarchical convolutional technology is the core of image network technology, which can extract network images’ local state feature information. The function of the pooling layer is to extract the global characteristics of the simulated image network by effectively reducing the input space dimension and the complexity of the image network structure. The network activation layer mainly provides nonlinearity for the network system model design and enhances the basic performance of modeling and generalization. The full connection layer maps and analyzes the network data category features obtained from the extracted data. As a data classifier, it puts them into the whole network data category, usually directly behind the whole network category structure.

### 2.3. Experiment Process

#### 2.3.1. Image Data Preprocessing

Figure 1a shows the original image of Arabidopsis with a flowerpot background, with a resolution of 320 × 320. Then, the LAB color space was used to segment the Arabidopsis image, as shown in Figure 1b. The L, A, and B components of the LAB color space can represent other colors, and the most major feature was that the brightness and color were not in the same channel. The L component only had a lightness, and the a and b components only had color. In the a and b components, the object’s outline was clear, and the LAB color space had a strong ability to distinguish color differences. Therefore, the equipment requirements for capturing images were relatively low, but it contained all color information in RGB mode. Figure 1c shows the segmentation mask region created on the original image. The yellow area is the image mask area obtained by segmentation using the LAB color space. Finally, the original image was segmented according to the mask region, and the Arabidopsis plant region could be extracted, as shown in Figure 1d. The Arabidopsis plant segmentation can be obtained by multiplying the binarized image of the mask area and the original image (RGB format).

#### 2.3.2. Experimental Data Division

The dataset consisted of 25 Cvi strains, 24 Col-0 strains, 26 Ler-1 strains, and 22 Sf-2 strains, totaling 97 Arabidopsis plants. Time-series images of 97 Arabidopsis plants were collected for 22 days, and 2134 images were obtained. The original resolution of the Arabidopsis image was 320 × 320. As shown in Table 1, 70% of the images were used for training and 30% for validation. To ensure the fairness of the experiment, the data in the training set and test set were randomly divided.

The image data of Arabidopsis organelles are shown in Table 2. The original image resolution (1024 × 1024) was adjusted to 150 × 150 to meet the size of the model input layer. The chloroplast, mitochondria, and peroxisome categories each contained 300 images. The images of each category were divided according to a 7:3 ratio. There were 630 images in the training set and 270 images in the verification set.

#### 2.3.3. Multiple Output Model Structure

Figure 1e shows the multi-output network structure, and the input layer size of this model was 150 × 150. The first part of the model consists of Convolution, Batch Normalization, and ReLU blocks. The convolution contains 16 5-by-5 filters. This is followed by the residual structure form, which includes the Convolution, Batch Normalization, and ReLU blocks. The convolution layer contains 32 3-by-3 filters. A skip connection was implemented around the previous two blocks containing a Convolution–BatchNorm–ReLU block with 32 1-by-1 convolutions. The data flow was then merged using an addition layer.

The multi-output model branching function was two different types of tasks added on top of the full connection layer. The first task branch, which performed the function of classifying Arabidopsis accessions, consisted of a fully connected layer with four nodes and branches operated by a SoftMax function. The second branch had the ability to predict the growth state of Arabidopsis, and the response content of the fully connected layer was the predicted growth day. The activation function of the fully connected layer in the branch is the rectilinear unit (ReLU). Figure 1f shows a schematic diagram of the multi-output model, which can more intuitively show the data flow of the model.

#### 2.3.4. Network Training Parameter

The Arabidopsis image data contained two labels, the first label was the type of genotype, and the second label was the number of days of growth status of Arabidopsis. In order to meet the size of the network input layer, the image resolution was adjusted to 150 × 150. The number of training iterations was 500, the batch size of learning was 128, and the learning rate was 0.001. With each iteration, the model parameters were adjusted to achieve optimal performance. The validation set data were then tested on the trained model to measure the model’s performance more accurately. Image processing and model building were carried out on the MATLAB 2021b platform.

#### 2.3.5. Model Performance Metrics

In the experiment, the model’s accuracy, sensitivity, specificity, false positive rate (FPR), and balanced F score (F1Score) were used to evaluate the method’s performance. These evaluation methods can directly show the model’s performance and facilitate the comparison of different methods. The formulas for measuring indicators are (1) to (5).
(1)$$Accuracy=\frac{TP+TN}{TP+FP+TN+FN}$$
(2)$$Sensitivity=\frac{TP}{TP+FN}$$
(3)$$Specificity=\frac{TP}{TP+FP}$$
(4)$$FPR=\frac{FP}{FP+TN}$$
(5)$$F1Score=2\times \frac{sensitivity\times precision}{sensitivity+precision}$$

The true positive (TP) was the number of positive samples and the true negative (TN) was the number of negative samples that the model predicted correctly. The false positive (FP) was the number of misclassified negative samples as positive and false negative (FN) was the number of positive samples misclassified as negative.

In the experiment RMSE (6) and R^2^ (7) were introduced to measure model prediction ability.

True Arabidopsis growth condition days: $y_{1},y_{2},y_{3},\ldots ,y_{n}$.

Predicted growth days of Arabidopsis using multiple output model: ${\hat{y}}_{1},{\hat{y}}_{2},{\hat{y}}_{3},\ldots ,{\hat{y}}_{n}$.

The formulas are as follows:(6)$$RMSE=\sqrt{\frac{1}{n}\sum_{i=1}^{n}{\left({\hat{y}}_{i}-y_{i}\right)}^{2}}$$
(7)$$R^{2}=1-\frac{\sum_{i=1}^{n}{\left(y_{i}-{\hat{y}}_{i}\right)}^{2}}{\sum_{i=1}^{n}{\left(y_{i}-{\bar{y}}_{i}\right)}^{2}}$$

To better evaluate the performance of the model, we also introduced residual prediction deviation (RPD) (8) into the experimental measurement.
(8)$$RPD=\frac{\sqrt{\frac{1}{n}\sum_{i=1}^{n}{\left(y_{i}-\bar{y}\right)}^{2}}}{RMSE}$$

$\bar{y}$ is the predicted days average.

The confusion matrix is a special contingency table with two dimensions (actual and predicted), and both dimensions have the same set of categories. Each matrix column represents a predicted class instance, while each row represents an actual class instance. The confusion matrix is a great visualization tool to see if a model tends to confuse two different classes easily.

The boxplot is a statistical graph often used in data analysis, which can visually observe outliers. The advantages of boxplots are as follows: The presentation of the boxplot is not affected by outliers. The boxplot can accurately and stably depict the discrete distribution of the data.

## 3. Results

### 3.1. The Performance of Multi-Output Model Identification in Arabidopsis

Figure 1g shows a large-size image synthesized from 64 Arabidopsis images, which was convenient to display the detailed information of the images. To study which regions contained important information, we imported the Arabidopsis image data with background into the network model. Figure 1h shows the Arabidopsis image features visualized by using the Grad-CAM method [30]. In the Figure 1h, the areas with apparent brightness represented important features of the Arabidopsis images, while the dark areas did not contain important feature information. The experimental results have shown that the recognition performance of the multi-output model trained on images with background was better than that of the model with training image data without background. The model’s classification accuracy, F1Score, and FPR with the training set containing images with backgrounds were 99.92%, 99.85%, and 0.0051. The model regression predicted the 22-day growth state performance indexes of *Arabidopsis thaliana*: RMSE was 1.5631, R^2^ was 0.9377, and RPD was 3.8974. In addition, we applied this model to Arabidopsis image data with backgrounds removed, and the classification accuracy was 93.36%, F1Score was 86.62%, and FPR was 0.0444. For the regression prediction, RMSE was 2.4109, R^2^ was 0.8518, and RPD was 2.4697, as shown in Table 3. The recognition accuracy of the multi-output model trained on images with backgrounds outperforms the model trained on images with no background by 6.56%, the F1Score is 13.23% higher, and FPR is 0.0393 lower.

It was an unexpected result that the model trained on images with backgrounds removed performed worse than the model trained on images with background. This counter-intuitive result challenges the common assumption that removing the background always improves model performance by allowing the model to focus on relevant features. Grad-CAM’s insight shows that even with the background, the model’s focus is still primarily on plant regions in the image. This observation is critical because it helps us to understand how the model processes image information. Figure 1i shows the confusion matrix of the model classification results of images with background. The model had an excellent classification effect on the image dataset with background, and there was only ambiguity in the LER-1 and COL-0 varieties. Figure 1j shows the boxplots of the 22-day growth state of *Arabidopsis thaliana* predicted by the model. The center of the boxplots was close to 0, which meant that the error margin for the predicted days was small. From the results in Figure 1j, it can be seen that the errors on the 2nd, 3rd, and 22nd days were larger. The predicted growth days of the Arabidopsis image are shown in Figure 1k. The green dotted line represents the day labels displayed, and the red dotted line represents the day labels predicted by the model. The smaller the angle between the two dashed lines, the smaller the error between the predicted result and the true label of the image. Figure 1l shows the confusion matrix of model classification results with image background removed. From the results in Figure 1l, the multi-output model had the best results in identifying the Arabidopsis lines (ler-1). The model was easily confused when identifying the two categories of col-0 and Sf-2. Figure 1m shows the boxplots of the 22-day growth state of *Arabidopsis thaliana* predicted by the model. It can be seen intuitively that the model performance without image background was far inferior to that with image background. Figure 1n shows the visualization of the prediction results. Based on the above results, the model’s performance in identifying Arabidopsis image data with background was better than with the background removed. In order to make the gap between the predicted result and the real result clearer and more intuitive, four images were selected and put together in the background with image and without image, respectively, as shown in Figure 2.

### 3.2. The Performance of Multi-Output Model Identification in Arabidopsis

Figure 3a shows the workflow of Experiment 2. Based on Experiment 1, the optimal model was selected and then these groups of data at different intervals were imported into this model for training. Image data were collected at intervals of 1, 2, 3, 4, and 5 days based on the original time-series images. Finally, we compared the performance of the models on datasets with different intervals. Figure 3b was a schematic flow chart of the multi-output model. The day interval sampling form is shown in the image selection method on the left side of Figure 3b, and then different datasets were imported into the model for training.

For the Task 1 model performance of images separated by one day, the classification accuracy was 98.97%, the F1Score was 97.91%, and the FPR was 0.0067. The regression prediction performance of Task 1 was as follows: RMSE was 2.0202, R^2^ was 0.8962, and RPD was 2.7532. The training time for Task 1 was 356 s. Figure 3c shows the confusion matrix for the model classification at 1-day intervals, and Figure 3d shows the boxplots of the predicted days for this interval group. The model had larger prediction errors on days 1 and 21, and outliers were predicted on day 3.

For the Task 2 model performance of images separated by two days, the classification accuracy was 98.55%, the F1Score was 97.17%, and the FPR was 0.0098. The regression prediction performance of the task was as follows: RMSE was 2.0202, R^2^ was 0.8962, and RPD was 2.7532. The training time for Task 2 was 261 s. Figure 3e shows the confusion matrix for the model classification at 2-day intervals, and Figure 3f shows the boxplots of the predicted days for this time interval group. The model’s prediction error was relatively large on days 4, 16, and 22.

For the model performance of Task 3 with images separated by three days, the classification accuracy was 97.43%, the F1Score was 94.93%, and the FPR was 0.0168. The regression prediction performance of the task was as follows, RMSE was 2.9951, R^2^ was 0.8135, and RPD was 1.8728. The training time for Task 3 was 204 s. Figure 3g shows the confusion matrix for the model classification at 3-day intervals, and Figure 3h shows the boxplots of the predicted days for that group. The model’s error was large on the 9th, 17th, and 21st days, and those errors were mainly manifested in that the number of predicted days were less than the actual number of days.

The model performance of Task 4 with images separated by four days was as follows: the classification accuracy was 94.88%, the F1Score was 90.01%, and the FPR was 0.0343. The regression prediction performance of the task was as follows: RMSE was 2.7598, R^2^ was 0.8374, and RPD was 2.1739. The training time for Task 4 was 159 s. Figure 3i shows the confusion matrix for the model classification at 4-day intervals, and Figure 3j shows the boxplots of the predicted days for that group. The error margin of the model was averaged over these five days.

The model performance of Task 5 with images separated by five days was as follows: the classification accuracy was 96.00%, the F1Score was 91.98%, and the FPR was 0.0274. The regression prediction performance of the task was as follows: RMSE was 2.5517, R^2^ was 0.8441, and RPD was 2.1249. The training time for Task 2 was 127 s. Figure 3k shows the confusion matrix for the model classification at 5-day intervals, and Figure 3l shows the boxplots of the predicted days for that group. The model had larger prediction errors on days 1 and 22.

A more intuitive comparison of several models can be made from Table 3. Task 1 has the best performance compared to Task 2, Task 3, Task 4, and Task 5. Task 1 has 4.09% higher classification accuracy, 7.9% higher F1Score, and 0.0276 lower FPR compared to the lowest, that of Task 4. Figure 3m shows the histogram of the classification accuracy of the models with different interval days. The classification accuracy of the model decreased as the interval increased. Figure 3n shows the RMSE histogram of model regression prediction. The model regression prediction ability decreased at intervals of 1 to 3 days, while the model regression prediction ability increased at intervals of 4 and 5 days. Figure 3o shows the training time of different tasks, and it can be seen intuitively that the training time of the model decreased with the increase of the interval days.

### 3.3. Performance of Multiple Output Models in Organelle Recognition

The purpose of this experiment was to demonstrate the robustness and generalizability of the model. Figure 4a shows the acquisition process for collecting Arabidopsis organelle images. Three types of Arabidopsis organelle images can be collected by electron microscope. Figure 4b shows a modified network structure based on a multi-task model, specifically retaining the model’s classification function and prohibiting the model’s regression prediction function. Figure 4c shows the Grad-CAM method used to visualize the image features of the model, and the visual image brightness area represents the key features of the image. The model learning rate was 0.001 and the number of training iterations was 650. Figure 4d shows the training process accuracy of this model; the final training accuracy was 98.56%. Figure 4e shows the loss change process of model training, and the final training loss was 0.026. Figure 4f shows the confusion matrix result of classification on the validation set, and the accuracy of the validation set was 99.25%.

## 4. Discussion

### 4.1. Advantages of the Multiple-Output Model

The multi-output model used in this paper was a kind of multi-label learning, which belongs to the category of multi-task learning [19]. Multi-task learning (MTL) has achieved great success in speech recognition and computer vision [31]. MTL improves the model effect by sharing the correlation information between multiple tasks [32]. Yasrab, R et al. [33] proposed a network model for the predictive segmentation of leaf and root appearance at a future time based on the time-series data of plant growth. The results showed a high degree of consistency and similarity between the predicted leaf frames and the ground truth, with the average SSIM for shoots being higher than 94.60% and the average mIoU for roots being higher than 76.89%. Chen et al.’s [34] study developed an effective attention-based neural network model to recognize Arabidopsis enhancers. The performance of the model on the test set was 0.955 for Mcc, 0.638 for AUPRC, and 0.837 for AUROC. However, in the above experimental analysis methods, the network models are only realized by a single task, whereas the present experimental model can obtain the results of multiple tasks at the same time using a single training dataset: *Arabidopsis thaliana* for classification as well as high-precision prediction of the growth state. This method can reduce two network structures to one network structure. This can not only save the computing power of the computer, but also make the network parameters smaller. It can also improve data efficiency and provide significant time savings. Overfitting can also be reduced by two different types of tasks sharing a single network structure. It reduces manual involvement and simplifies the design and training process of the model. Moreover, the recognition accuracy of this experimental model is 99.28%, and the root mean square error of growth state regression prediction is 1.536. The model proposed in this paper can not only complete the classification, but also complete the regression prediction task. In Experiment 2, examining the performance of multiple output models at different time intervals compared to each other, the model gradually reduced the training time. In Experiment 3, the multi-output model was modified into a single-task learning mode. At the same time, the model proposed in this paper can continue to add learning tasks, which provides great help for the transformation of model application scenarios. As a result, we can choose the learning mode according to the research content in the future.

### 4.2. Effects of Images with and without Background on Model Performance

The purpose of removing the image background is to let the model focus only on plant information. However, the experimental results show that the performance of the model with the background removed is not as good as the performance of the model with the background. Using Grad-CAM, the recognition area of the key features of the image can be visually observed [35]. It can be seen that the key information of the model to the image was still concentrated in the image plant area. We hypothesize that the influencing factors for the poor performance of the preprocessed image model are as follows: Influencing factor 1—image preprocessing can negatively affect the plant edge information, i.e., too low an image resolution may result in the plant leaf information not being segmented correctly during the preprocessing operation. The above operation may damage the key feature information of the plant. Influencing factor 2—the reason for the superior performance of the model in recognizing Arabidopsis images with background may be due to the fact that the background information is meaningful to the model performance. In observing the soil background, it can be seen that the background color of the soil gradually deepens with time. Subsequently, this influencing factor needs to be verified by designing experiments, and we intend to remove the plants, leaving only the soil background, and import the soil background into the network model to further test the soil background on the model recognition performance. Additionally, there are some differences in the Arabidopsis seeds themselves, which will have some effects on individuals and may lead to slight fluctuations in the accuracy of the experiment. However, judging from the indication of the overall trend, the present experimental model and method are still very reliable.

### 4.3. The Effect of Increasing Interval Days on the Model

To prove the robustness of the model proposed in this paper, we adopt the method of reducing the time-series image data. It can be known from the experimental results that increasing the collection gap of the collected images leads to a general decline in the performance of the model classification, while the regression prediction ability remains relatively stable. The reasons for the degradation in classification performance are as follows:

First, the expanded time gap in the image dataset resulted in less training image data, and as a result, the model could not undergo sufficient training iterations of the images [36]. In addition, the increase of the image time interval increases the differences between the images, so it was very difficult for the model to recognize subsequent images [37]. Larger time intervals result in less image data being acquired, which leads to lower classification performance of the model. When the time interval is from 1 to 4 days, the accuracy decreases significantly as shown in Figure 3m. However, the accuracy rate increases when the time interval is 5 days, which may be due to the phenomenon of overfitting that occurs when too few images are trained. External factors could also be contributing factors. In the time interval of 5 days, *Arabidopsis thaliana* may be affected by insect bites, soil moisture, and other factors that lead to the plant itself producing a more obvious difference, which leads to the model classification accuracy being increased.

When the regression prediction was expanded to datasets with the time gap, the main reason for the more stable prediction ability may be that there were fewer model regression prediction categories, so the difficulty of regression prediction was reduced. In terms of model structure, the convolutional neural network used in the model is based on weight sharing and local linking, which reduces the number of weights and complexity of the model collection. The pooling layer in the CNN effectively reduces the input spatial dimensionality and complexity of the image network structure and extracts the global features of the simulated image network. The use of skip connections and additional layers helps to improve the error and reverse error paths, which all contribute to the stability of the model regression performance. In the overall view of a decreasing trend, the results are still illustrative. In future work, the number of samples of experimental materials should be increased, and Arabidopsis lines should be expanded to reduce the experimental error.

### 4.4. Effect of Fluorescence on Image Visualization

When using fluorescent images to visualize plant cells, there are also factors that can affect what can be observed. In terms of the specificity of the fluorescent marker, the fluorescently labelled molecule or protein must bind or localize to the target cell structure with a high degree of specificity [38]. The specificity of the labeling directly affects the clarity and accuracy of detail in the image. Non-specific binding or drift of the labeling may result in blurred images or incorrect interpretation. Fluorescence intensity depends on the concentration of the fluorescent dye, the intensity of the excitation light, and the expression level of the organelle or protein [39]. Insufficient fluorescence intensity may lead to a loss of detail, while too much intensity may lead to overexposure and a loss of image detail. Autofluorescence and background signals will affect the detail of the visualization to some extent [40]. Some plant cellular components naturally emit fluorescence, which may mix with added fluorescent labeling signals and affect image interpretation. Background signals, both autofluorescence and non-specific fluorescence, can reduce the contrast and clarity of an image. When using multiple fluorescent markers to simultaneously label different cellular structures or proteins, wavelength selection and fluorescence overlap can make image interpretation difficult.

### 4.5. Discussion of Image Data and Deep Learning

Deep learning is a learning method based on a large amount of data, so we need a large amount of image data to support model training and validation [41]. The 22-day growth image of *Arabidopsis thaliana* was selected in this experiment because of the limitations of the experimental setting and the growth cycle of the plants used. To verify the algorithm’s ability to classify growth stages (accurate to the day), the dataset included time-delayed video recordings of Arabidopsis varieties, where images were taken at a specific time (12 noon) each day. The 22-day cycle covers *Arabidopsis* from seedling to maturity and is an important stage in the plant growth cycle of interest to researchers. In the trend of using deep learning technology to analyze plants, it is very important to refine the content of research, including predicting plants every day or even every hour, which has certain research value. This cycle also ensures the manageability of the data, both in terms of the number of images to be processed and the computational resources required to analyze the time-series with deep learning methods [25]. Chen et al. [42] reviewed the methods of deep learning for plant image recognition in the past five years focusing on the structure of the convolutional neural networks used in plant recognition and classification and the methods of improving convolutional neural networks, as well as the ways of image collection and processing. Picek et al. [43] proposed a novel image retrieval method based on nearest neighbor classification, which can be carried out in deep embedded space, providing a new solution for complex recognition tasks. This method can not only improve the accuracy and robustness of classification, but also help the visualization of prediction samples, and achieve good results.

### 4.6. Discussion of Deep Learning Models

Zhou et al. [44] used six deep learning network models to test the classification performance on four publicly available plant datasets and a self-made camellia dataset (Camellia@clab). The prediction performance was evaluated by 10 times 10-fold cross-validation. This work is a preliminary study applying deep learning models to plant image classification, and in future work, more factors such as biometrics and data can be fused together as features. Gao et al. [45] proposed some pioneering deep learning models to integrate these big data and stimulate new deep learning multimodal data fusion techniques. Some of the challenges and future topics of deep learning models for multimodal data fusion are described. Huixian et al. [46] proposed an artificial neural network classification method based on the backpropagation error algorithm (BP algorithm) to identify plant leaves. The accuracy of classification of the seventh blade of this model reached 92.48%. It proves that using deep learning to classify the leaves of plants is feasible. The convolutional neural network used in this experiment is somewhat different from the artificial neural network. Suherman et al. [47] achieved 98.2 percent accuracy in rice classification training using artificial neural networks and 99.3 percent accuracy using convolutional neural networks. In order to pursue a higher accuracy, the convolutional neural network is chosen.

### 4.7. Future Work

In this study, the model was a 2-output model, but the actual application situation may require more task learning. Therefore, we need to transform multiple models to deal with the actual situations and verify them in practical applications to prove the model constructability. There are already several applications including Pl@ntNet, Plant-classifier, Leaf-Image-Classification, Flora Incognita, and other apps that can classify plants on your smartphone. However, these programs can only identify plant species, not different lines of the same species.

Machine vision technology can provide high-throughput index sets much higher than conventional agronomic traits, providing more ways to analyze the relationship between various agronomic traits and yield path analysis [48,49,50]. However, it also results in high-dimensional index sets and valuable technical difficulties in information extraction.

This experiment still has some shortcomings as the experiment was not tested on other plants. Since *Arabidopsis thaliana* is the model plant, we verified the feasibility of this method on it first. In Experiment 3, we used transfer learning to test the model on organelles. The model was found to perform well, confirming the reliability of the method. Eventually, we have to design a complete system that can be analyzed for the whole plant. In future studies, we will also be sure to test the model on different plants to further validate the robustness of the model. The size and location of organelles is also very important information in biology and is well worth further study. In our next experiment, we will also use target detection methods (e.g., YOLO, R-CNN, etc.) to further determine the size and location of organelles. This method also contributes to the phenotypic measurements and helps to promote the improvement of the comprehensive information of plant phenotypes.

## 5. Conclusions

The morphological structure of plants has important biological significance in agricultural production and research. At this stage, it is crucial to use machine vision to identify and monitor the morphological structure of plants. This paper proposes a multi-task learning method to complete the image genotype classification and growth status prediction of *Arabidopsis thaliana*. In the classification problem, the recognition accuracy of the model was 99.28%, and the RMSE of regression prediction of growth status was 1.536. The experimental results show that this method is excellent at identifying plant categories and can stably predict the plant growth status. *Arabidopsis thaliana* is an important model plant in basic biology and is significant in agricultural research, so this study has general implications for other crop studies. This method provides technical support for developing a high-throughput information collection platform for plants and helps in dealing with more complex situations.

## Figures and Tables

**Figure 1 plants-13-01177-f001:**
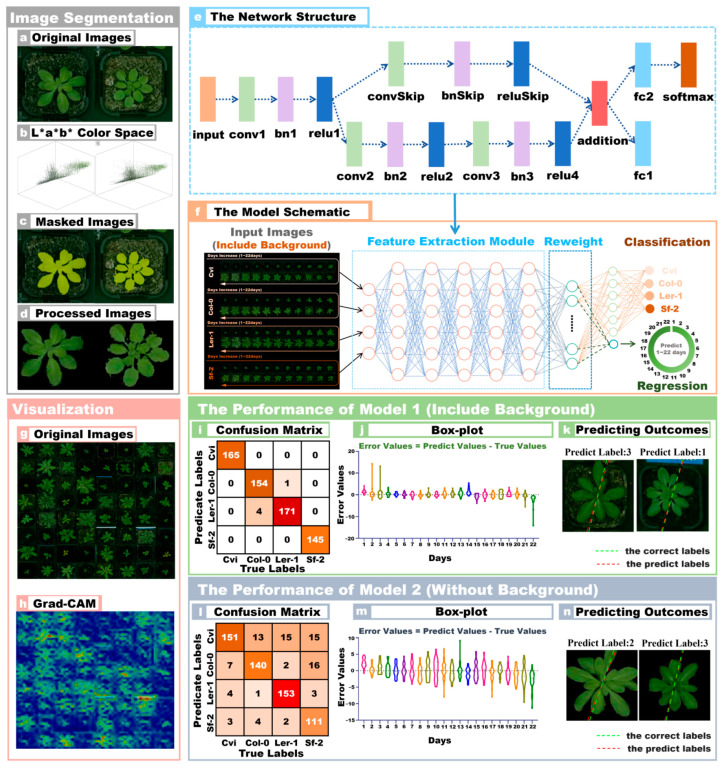
Multi-output model performance in Arabidopsis images. (**a**) The original images of *Arabidopsis thaliana*; (**b**) the image components in LAB color space; (**c**) the masked images; (**d**) the segmented images; (**e**) the network model structure; (**f**) the schematic diagram of network model; (**g**) the original images of the input model; (**h**) the visualized image features using Grad-CAM; (**i**) the confusion matrix of the model on the images with background; (**j**) the boxplot of the model on the images with background; (**k**) the model predictions on the images with backgrounds; (**l**) the confusion matrix of the model on the images with backgrounds removed; (**m**) the boxplots of the model on the images with backgrounds removed; (**n**) the predictions of the model on the images with backgrounds removed.

**Figure 2 plants-13-01177-f002:**
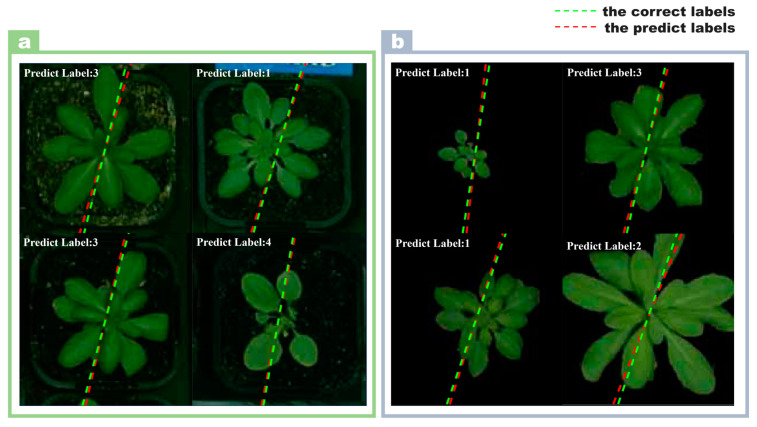
Prediction results of Arabidopsis images by multi-output model. (**a**) The model predictions on images with backgrounds; (**b**) the predictions of the model on images with backgrounds removed.

**Figure 3 plants-13-01177-f003:**
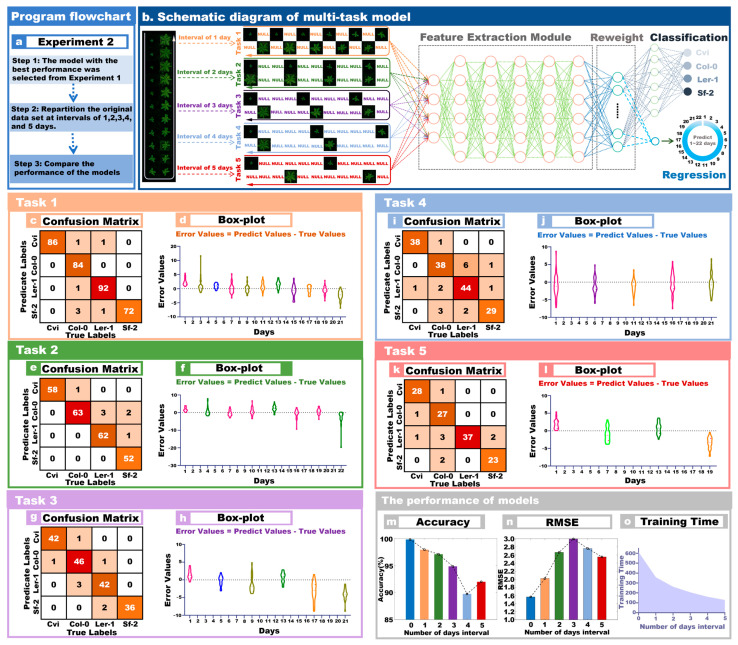
Model performance on different day gaps. (**a**) The flow of the experiment; (**b**) the schematic diagram of the multi-output model on different day gaps; (**c**) the confusion matrix of the 1-day gap model; (**d**) the boxplot of the 1-day gap model; (**e**) the confusion matrix of the 2-day gap models; (**f**) the boxplots for models with 2-day intervals; (**g**) the confusion matrix for models with 3-days intervals; (**h**) the boxplots for models with 3-day intervals; (**i**) the confusion matrix for models with 4-day intervals; (**j**) the boxplots for models with 4-day intervals; (**k**) the confusion matrix for models with 5-day intervals; (**l**) the boxplots for models with 5-day intervals; (**m**) the classification accuracy of models at different time intervals; (**n**) the predicted RMSE of the model at different time intervals; (**o**) the training time of the model at different interval times.

**Figure 4 plants-13-01177-f004:**
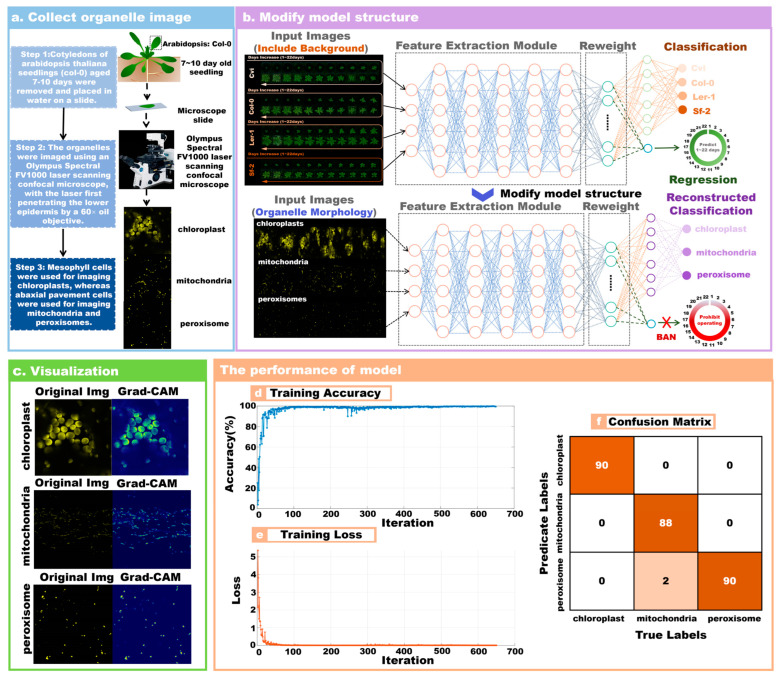
The performance results of the improved network model for the classification of Arabidopsis organelles. (**a**) Experimental workflow for collecting images of Arabidopsis organelles; (**b**) modifying the structure of the model; (**c**) visualizing image features using Grad-CAM; (**d**) accuracy curves for iterative training pairs of the model; (**e**) the loss curve of iterative training pairs; (**f**) the confusion matrix of the modified model.

**Table 1 plants-13-01177-t001:** The detailed information of Arabidopsis images.

Category	Resolution	Training Set	Validation Set	Total
Cvi	320 × 320	385	165	550
Col-0	320 × 320	370	158	528
Ler-1	320 × 320	400	172	572
Sf-2	320 × 320	339	145	484

**Table 2 plants-13-01177-t002:** The detailed information of Arabidopsis organelle images.

Category	Resolution	Training Set	Validation Set	Total
chloroplast	1024 × 1024	210	90	300
mitochondria	1024 × 1024	210	90	300
peroxisome	1024 × 1024	210	90	300

**Table 3 plants-13-01177-t003:** The performance of the multi-output model.

Model	Classification			Regression			Training Time
	Accuracy	F1Score	FPR	RMSE	R^2^	RPD	
Model 1	99.92	99.85	0.0051	1.5631	0.9377	3.8974	3676
Model 2	93.36	86.62	0.0444	2.4109	0.8518	2.4697	3253
Task 1	98.97	97.91	0.0067	2.0202	0.8962	2.7532	356
Task 2	98.55	97.17	0.0098	2.6659	0.8389	2.4060	261
Task 3	97.43	94.93	0.0168	2.9951	0.8135	1.8728	204
Task 4	94.88	90.01	0.0343	2.7598	0.8374	2.1739	159
Task 5	96.00	91.98	0.0274	2.5517	0.8441	2.1249	127

## Data Availability

Data is publicly available at: http://phenocam.anu.edu.au/cloud/a_data/_webroot/published-data/2017/2017-Namin-et-al-DeepPheno.zip (accessed on 16 April 2024).

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
