# Peer review of "From Organelle Morphology to Whole-Plant Phenotyping: A Phenotypic Detection Method Based on Deep Learning"

_plants, 2024, doi:10.3390/plants13091177_

Round 1
Reviewer 1 Report (New Reviewer)
Comments and Suggestions for Authors
This article is very interesting for plant phenotyping. I have found it very useful for characterizing plants for breeding work.
Suggestions for Autors
Line 15 Arabidopsis thaliana change in Arabidopsis thaliana
Line 17 I would use the term accessions instead of genotypes
Line 49 Gebhardt et al. change in Gebhardt et al.
Line 51 Abbasgholipour et al. change in Abbasgholipour et al.
Line 66 Azim et al. change in Azim et al.
Line 69 Saleem et al. change in Saleem et al.
Line 77 Giuffrida et al. change in Giuffrida et al.
Line 79 Lim et al. change in Lim et al.
Line 80 Kolhar et al. change in Kolhar et al.
Line 83 Mishra et al. change in Mishra et al.
Line 85 Arabidopsis thaliana change in Arabidopsis thaliana
Line 85 Xu et al. change in Xu et al.
Line 92 Dobrescu et al. change in Dobrescu et al.
Line 94 Wang et al. change in Wang et al.
Line 106-107 These acronyms (GPUs and TPUs) should be spelled out in full the first time they are mentioned
Line 111 between tasks cahnge in among tasks
Line 112 between tasks cahnge in among tasks
Line 118 Arabidopsis thaliana change in Arabidopsis thaliana
line 121 arabidopsis thaliana change in Arabidopsis thaliana
Lines 123 and 124 Arabidopsis thaliana change in Arabidopsis thaliana
Line 129 arabidopsis change in Arabidopsis
Lines 131 132 134 arabidopsis thaliana change in Arabidopsis thaliana
Lines 137/138 Arabidopsis thaliana change in Arabidopsis thaliana
Line 144 arabidopsis change in Arabidopsis
Line 144 I would use the term accessions instead of genotypes
Line 144 What do the acronyms Sf-2, Cvi mean?
Lines 147-148 Arabidopsis thaliana change in Arabidopsis thaliana
Lines 157-158 arabidopsis change in Arabidopsis
Lines 192-197-202-208 -220- 227 arabidopsis change in Arabidopsis
line 220 I would use the term accessions instead of genotypes
Line 228 in this case if it is a single plant the term genotype can be used
Line 236 What do the acronyms FPR, and F1Score mean?
Line 282 Arabidopsis thaliana thaliana change in Arabidopsis thaliana
line 284 arabidopsis change in Arabidopsis
Line 310 Arabidopsis thaliana thaliana change in Arabidopsis thaliana
Line 313 arabidopsis change in Arabidopsis
Lines 316-317 Arabidopsis thaliana thaliana change in Arabidopsis thaliana
Line 470 Arabidopsis thaliana thaliana change in Arabidopsis thaliana
Line 496 Arabidopsis thaliana thaliana change in Arabidopsis thaliana
Line 507 Arabidopsis thaliana thaliana change in Arabidopsis thaliana
Line 510 Arabidopsis thaliana thaliana change in Arabidopsis thaliana
Author Response
Please see the attachment.

Reviewer 2 Report (New Reviewer)
Comments and Suggestions for Authors
Review of the manuscript entitled: ‘Organelle morphology to whole-plant phenotyping: a phenotypic detection method based on deep learning’ by Liu et al, submitted to the Plants
This work concerns the possibility of the analysis of plant phenotype parameters using deep learning, trying to persuade that it is possible to classify Arabidopsis thaliana from the macro (plant) to the micro-level (organelle) using this technique.
This work is quite well well-written, original, novel study, however, I have some doubts and I have listed specific points/questions that require attention by the Authors. Based on these data and method description I am not convinced that all biological part was set up properly. Probably there is a need to discuss data with plant specialists – botanist and cell biologist. Only some doubts are listed below, which makes it extremely difficult to assess the quality and reliability of the data generated when such fundamental errors are made in describing species names and methods.
1. Lines 158-160 – Authors claim that they used electron microscopy. Seriously? SEM or TEM? There is a lack of appropriate method description for visualization of organelles, equipment details, setup
2. From Fig.3- I can assume that there were no electron microscopy techniques used – see previous remark – as it is written in lines 158-160 and 392-393. These images are from confocal microscopy (CLSM, as I think), based on fluorescent images!!! All details are necessary that can influence visualization
3. Given the above how Authors were able to visualize organelles – for chloroplasts, it is easy based on autofluorescence, however, it is not clear which detection method was used for peroxisomes and mitochondria. Was it immunolocalization or another technique? All details MUST be provided
4. There is a need to check through the text species name spelling rules, lots of errors are there e.g. line 15, 123 etc – no italic, line 121, 130, 131 etc – ‘arabidopsis thaliana’ instead of Arabidopsis thaliana. It is commonly known that the scientific names of species are italicized. The genus name is always capitalized. Other rules are completely unacceptable in scientific papers
5. What does it mean: ‘organelle species’ – line 126?
6. What does it mean ‘organelle performance’ – line 403?
7. Line 118 and 510 - ‘model plant for botanical and crop studies’ – crops are also plants (thus botanical), do you mean agricultural studies?
8. Line 208 – ‘organel images’?
9. Line 497 – ‘In Experiment 3, we used transfer learning to test the organelles’ – For which parameters – only presence or also differences in size, shape, localization within cells etc.? Only for presence - it does not make any sense, we have already known this…
Summarizing, it is not easy to make a final conclusion that it is possible ‘to classify Arabidopsis thaliana from the macro (plant) to the micro-level (organelle) using this technique’ (comment: it is not ultrastructural level)
Comments on the Quality of English Language.
Author Response
Please see the attachment.

Reviewer 3 Report (New Reviewer)
Comments and Suggestions for Authors
The article “Organelle morphology to whole-plant phenotyping: a phenotypic detection method based on deep learning” presents a deep learning multi-output model to classify Arabidopsis thaliana genotype species. The model was also applied to classify Arabidopsis organelles.
The article's findings have merit, and the principles of a multi-task learning method are sufficiently clear. However, in the abstract, the authors should emphasize the novelty of the research. In the discussion section, the use and role of this type of machine learning method for studying plant phenotypes should be widely discussed.
The text is well organized. However, the authors must address some minor issues, as presented below.
1. All abbreviations, such as FPR and RPD, must be explained.
2. R must be squared instead of R2.
3. Figures 1k and 1n – the dotted lines in the images are not apparent. Please provide a clearer image.
4. The materials and methods do not describe the method of obtaining images of cell organelles and preparing these images. The concerns about this issue are included only in Figure 3a. It is too little and too far in the text.
Round 2
Reviewer 2 Report (New Reviewer)
Comments and Suggestions for Authors
Authors have made substantial changes and modification that, as I believe, can significantly strenghten the paper. However I still have a doubt:
1. In cell biology, we do not use the term "organelle species", but simply organelles, which means different types of organelles: mitochondria, plastids etc
Comments on the Quality of English Language
This ms was improved in the term of my previous remarks
Author Response
To editor and reviewers:
I sincerely appreciate your feedback on the manuscript. Based on your comments, we have made revisions to the relevant sections and adjusted the wording throughout the paper.
Thank you for your valuable input, and we believe these changes have strengthened the quality of the manuscript.
Response to Reviewer 2 Comments
Authors have made substantial changes and modification that, as I believe, can significantly strenghten the paper. However I still have a doubt:
Point 1: In cell biology, we do not use the term "organelle species", but simply organelles, which means different types of organelles: mitochondria, plastids etc
Response 1: We have modified this part, and we use organelles instead of the term "organelle species". Thank you! All changes are highlighted in red.
Page 3, Line 126-128: Identification of Arabidopsis organelles (chloroplast, mitochondria and peroxisome) using an improved multitasking model to test the migratory nature of the method.
Page 4, Line 156-157: To verify the generalization of the proposed method, we also applied the model to identify Arabidopsis organelles[23].

This manuscript is a resubmission of an earlier submission. The following is a list of the peer review reports and author responses from that submission.
Round 1
Reviewer 1 Report
Comments and Suggestions for Authors
The paper authored by Liu et al. presents a comprehensive exploration of deep learning applications in plant phenotyping, addressing the classification of Arabidopsis at various scales and incorporating multi-task learning for genotype classification and growth status prediction. The study delves into the influences of image background and time intervals on model performance, offering valuable insights into the strengths and challenges of the proposed methodology. The novelty of the approach and the significance of the discussed topic render it a potentially valuable addition to the plant phenotyping domain. The Methods and Materials section covers dataset particulars, CNN architecture, and an extensive experimental process, encompassing image preprocessing, data division, model structure, training parameters, and performance metrics for a thorough evaluation.
Given the importance of the proposed methodology and the insights gleaned from the study, I recommend the manuscript for major revision. Addressing the following points will enhance the paper's quality and contribute to its overall impact:
-
1- Authors are asked to provid additional information about the validation dataset, the experimental setup, and any challenges encountered during validation.
-
2- Authirs should strengthen the outcomes contribution by conducting a more comprehensive comparison with existing methods in plant phenotyping. A discussion on how the proposed approach compares to or improves upon these methods is crucial.
-
3- Please Elaborate on the unexpected finding that model performance with the background was superior. Provide a detailed explanation, addressing potential causes and presenting viable solutions to clarify this intriguing outcome.
-
4- Authors should conduct a more in-depth analysis of factors contributing to reduced classification performance with larger time intervals. Propose potential strategies for mitigating these challenges to enhance the paper's depth.
-
5- Authors s must discuss whether the approach has been tested in other plants, such as Monocots/Dicots, C3, C4, and CAM. Anticipate how authors envision the application of the approach in these different plant categories.
-
6- While the figures are well-designed, there is concern reqarding their quality and readability. Ensure that figures are presented clearly at an appropriate resolution to facilitate effective communication.
Minor editing of English language required
Reviewer 2 Report
Comments and Suggestions for Authors
This paper employs multi-output deep learning methods for plants phenotyping and growth prediction based on computer vision. With trending applications of computer vision and machine learning in precision horticulture, this paper describes smart methodologies to enhance our understanding of plants' growth. It is well within the scope of the Plants Journal and interesting.
The authors have well cited relevant literature throughout the paper. The paper is generally well read and informative. The method has been clearly described.
However, I am struggling with the novelty and contribution of the methods described in this paper, especially in comparison with Taghavi Namin et. al 2018, from which the authors got the data. The multi-output multi-task deep learning methods have already been applied to different datasets as the authors have cited. So, I read it as applying some existing methods on some existing dataset. This is fine as long as we are mindful of the audience. If the target audience are machine learning experts, I wonder if the methodological novelty would be sufficient to publish. However, if the target audience are plant breeders and physiologists (which I believe is the case), the focus of the paper should be more around the conclusions and applications. The paper would benefit significantly from some rearrangements, moving technical details into the appendix, and more elaboration on how the method could enhance knowledge in the application domain.
To be more specific:
1. The abstract reports on the three models trained and validated in the paper, but not much on the "why" factor. It could be revised based on the points made in the introduction and discussion sections.
2. Line 54 in the introduction claims that the performance of image processing methods was not stable. It needs more elaboration and possibly citation.
3. Please elaborate what you mean by completing classification. Does it mean higher precision?
4. The paragraph starts in line 87 of the introduction, claims limitations in the multiple classification models. What are those limitations?
5. Section 2.3.5 can be in an appendix, as the metrics definitions are standard. They could just be mentioned alongside with the models' factors and software used.
6. The text in the results (section 3) mainly repeats the numbers in the related tables. It can be confusing for the reader to follow what the numbers mean in practice. Most importantly, due to the authors' claim of the models' novelty, it makes sense to compare the performance with the previous single-task methods in the results section. For example, bringing evidence that the classification accuracy or method precision with an existing model (either cited or applied by the authors in this paper) was x% lower than the proposed model.
7. Same goes with the claim in section 4.1. about shortening the training time. Please provide comparison between the proposed models' training times versus the existing work. If shortening the training time is part of the contribution of the paper, it should also be mentioned in the introduction.
8. The results and conclusions on the regression model need more elaboration. The model's performance (RMSE) was reported, joined with a small graph on the right-hand side of fig 1 to 3. However, since the regression model was part of the contribution, it needs more attention. Figures with predicted vs observed growth for a few plants would be helpful. In addition, section 4.3 needs more elaboration on the regression performance and stability. The explanation on the stability of the regression model was not convincing as it seems to be more due to the data than the model.
9. Section 4.2 makes some good points that could be more highlighted in the results section.
Some minor comments:
a. When you use an abbreviation for the first time in the paper, use the complete names. For example, GAHSI in line 54, or KNN in line 60.
b. The references in line 68 (Kohler at al. Lim et al.) doesn't seem right.
c. The quality of figures could be improved. Some texts are too small to read.
Comments on the Quality of English LanguageGood comprehensive English in general
Round 2
Reviewer 1 Report
Comments and Suggestions for Authors
the authors have successfully addressed the concerns raised during the review process. Their diligent efforts have significantly improved the quality and clarity of the manuscript.
Perhaps before the final acceptance, the contents of the figures are not legible. Please replace the image with one of a sufficiently high resolution (min. 1000 pixels width/height, or a resolution of 300 dpi or higher).
Also, there are some minor errors that should be picked up on revision.
Comments on the Quality of English LanguageMinor editing of English language required
Reviewer 2 Report
Comments and Suggestions for Authors
Thank you for your time and effort addressing all the comments. I think the revised version reads well and comprehensive. The novelty of the work and the findings are nicely communicated. The discussion section makes very interesting points and comparisons to the previous work.